Accuracy of the interferon-gamma release assay for the diagnosis of tuberculous pleurisy: an updated meta-analysis

Pang Cai-Shuang
Shen Yong-Chun
Tian Pan-Wen
Zhu Jing
Feng Mei
Wan Chun
Wen Fu-Qiang wenfuqiang.scu@gmail.com
Department of Respiratory and Critical Care Medicine, West China Hospital of Sichuan University and Division of Pulmonary Diseases, State Key Laboratory of Biotherapy of China , China
Zuo Li
Electronic publication date: 2015 May 21
Publication date: 2015
Volume: 3
Electronic Location ID: e951
Received 2015 Mar 2; Accepted 2015 Apr 21
Copyright: © 2015 Pang et al.
Copyright year: 2015
Copyright holder: Pang et al.
License: This is an open access article distributed under the terms of the Creative Commons Attribution License, which permits unrestricted use, distribution, reproduction and adaptation in any medium and for any purpose provided that it is properly attributed. For attribution, the original author(s), title, publication source (PeerJ) and either DOI or URL of the article must be cited.
License URL: https://creativecommons.org/licenses/by/4.0/

Keywords: Interferon-gamma release assay, Tuberculous pleurisy, Diagnosis, Meta-analysis

Funding: National Natural Science Foundation of China 81230001 81300032 This work was supported by grants 81230001 and 81300032 from the National Natural Science Foundation of China. The funders had no role in study design, data collection and analysis, decision to publish, or preparation of the manuscript.

==============================
Background and Objectives. The best method for diagnosing tuberculous pleurisy (TP) remains controversial. Since a growing number of publications focus on the interferon-gamma release assay (IGRA), we meta-analyzed the available evidence on the overall diagnostic performance of IGRA applied to pleural fluid and peripheral blood.

Materials and Methods. PubMed and Embase were searched for relevant English papers up to October 31, 2014. Statistical analyses were performed using Stata and Meta-DiSc. Pooled sensitivity, specificity, positive likelihood ratio (PLR), negative likelihood ratio (NLR), positive predictive value (PPV), negative predictive value (NPV) and diagnostic odds ratio (DOR) were count. Summary receiver operating characteristic curves and area under the curve (AUC) were used to summarize the overall diagnostic performance.

Results. Fifteen publications met our inclusion criteria and were included in the meta analysis. The following pooled estimates for diagnostic parameters of pleural IGRA were obtained: sensitivity, 0.82 (95% CI [0.79–0.85]); specificity, 0.87 (95% CI [0.84–0.90]); PLR, 4.94 (95% CI [2.60–9.39]); NLR, 0.22 (95% CI [0.13–0.38]); PPV, 0.91 (95% CI [0.85–0.96]); NPV, 0.79 (95% CI [0.71–0.85]); DOR, 28.37 (95% CI [10.53–76.40]); and AUC, 0.91. The corresponding estimates for blood IGRA were as follows: sensitivity, 0.80 (95% CI [0.76–0.83]); specificity, 0.70 (95% CI [0.65–0.75]); PLR, 2.48 (95% CI [1.95–3.17]); NLR, 0.30 (95% CI [0.24–0.37]); PPV, 0.79 (95% CI [0.60–0.87]); NPV, 0.75 (95% CI [0.62–0.83]); DOR, 9.96 (95% CI [6.02–16.48]); and AUC, 0.89.

Conclusions. This meta analysis suggested that pleural IGRA has potential for serving as a complementary method for diagnosing TP; however, its cost, high turn around time, and sub-optimal performance make it unsuitable as a stand-alone diagnostic tool. Better tests for the diagnosis of TP are required.

Introduction

Tuberculous pleurisy (TP) is the most common form of extrapulmonary tuberculosis, accounting for 23% of all tuberculosis cases and 30% of cases of disease-causing pleural effusion (PE) (Vidal et al., 1986; Corbett et al., 2003; Valdés et al., 2003), which involves exudate containing primarily lymphocytes. Direct diagnosis of TP would be the best way to avoid misdiagnosis and the resulting inappropriate treatment (Lin et al., 2009), but this remains a challenge. Definitive diagnosis of TP depends on isolating Mycobacterium tuberculosis from PE or pleural tissue. Conventional methods, such as PE culture, pleural biopsy and Ziehl-Neelsen staining, show poor sensitivity for detecting the limited amounts of bacteria in the PE of affected patients (Escudero et al., 1990; Valdés et al., 1998). Culturing PE is also time-consuming. Pleural biopsy is invasive and technically difficult, so its effectiveness depends on technical skill (Pérez & Jiménez, 2000). It may not be suitable for elderly and children, individuals with underlying co-morbidities, and those at high risk of bleeding. The tuberculin skin test is cross-reactive for Bacille Calmette Guérin (BCG) and many non-tuberculous mycobacteria, increasing the risk of misdiagnosis (Lawrence, 2000; Stead & To, 1987; Liebeschuetz et al., 2004). The limitations of these conventional approaches to diagnosing TP highlight the need to identify new diagnostic tools.

The PE of patients with TP has been shown to contain significantly higher levels of T lymphocytes and interferon (IFN)-γ than peripheral blood (North & Jung, 2004; Sharma et al., 2002), and the PE of these patients contains higher IFN-γ levels than the PE of uninfected individuals (Yamada et al., 2001). In fact, T lymphocytes that have previously been exposed to MTB release more IFN-γ on repeat exposure. This inspired the development of a T-cell IFN-γ release assay (IGRA), which is now licensed as a blood test for diagnosis of latent tuberculosis (Lalvani, 2007; Pai, Zwerling & Menzies, 2008).

Whether IGRA can be used to diagnose TP is controversial. A previous meta-analysis concluded that it showed poor sensitivity and specificity for this purpose (Zhou et al., 2011). Nevertheless, a growing number of studies have focused on extending the use of IGRA to the diagnosis of TP (Hooper, Lee & Maskell, 2009). Therefore, the present meta-analysis was undertaken to comprehensively assess the overall accuracy of IGRA for the diagnosis of TP.

Material & Methods

Search strategy and study selection

PubMed and Embase were searched for articles published before October 31, 2014. The following search terms were used: “pleural effusion/pleural fluid, pleurisy/pleuritis AND elispot, OR quantiferon, OR interferon-gamma assays, OR interferon-gamma release assays, OR t cell assays.” The related-articles function was also used, and reference lists in relevant articles were searched manually.

Studies were included in our meta-analysis if they (1) used IGRA testing for the diagnosis of tuberculous pleurisy (2) reported sufficient data to calculate true positive, false positive, false negative , and true negative of IGRA for the diagnosis of TP, and (3) constituted original research published in English. Studies available only as abstracts were excluded.

Data extraction and quality assessment

Two reviewers independently checked all potentially relevant studies, and disagreements were resolved by consensus. Data were collected from each study, including first author, year of publication, country, participant characteristics, IGRA method, samples, cut-off values, sensitivity, specificity and methodological quality. For each study we constructed 2 × 2 contingency tables in which we calculated true positive, false positive, false negative, and true negative rates.

The methodological quality of the studies was assessed using the 14-items Quality Assessment for Studies of Diagnostic Accuracy (QUADAS) guidelines (Whiting et al., 2003). When a criterion was fulfilled, a score of 1 was given, 0 if a criterion was unclear, and −1 if a criterion was not achieved. This evaluation instrument rates studies on a quality scale of up to 14 points.

Statistical analyses

Standard methods recommended for meta-analyses of diagnostic test evaluations (Devillé et al., 2002) were used. Stata 12.0 and Meta-DiSc 1.4 were used for statistical analysis. The following accuracy measures were calculated for each study: sensitivity, specificity, positive likelihood ratio (PLR), negative likelihood ratio (NLR), positive predictive value (PPV), negative predictive value (NPV) and diagnostic odds ratio (DOR). Summary receiver operating characteristic (SROC) curves and area under the curve (AUC) were also calculated (Moses, Shapiro & Littenberg, 1993; Irwig et al., 1995; Vamvakas, 1998). Heterogeneity across studies was detected using chi-square and Fisher’s exact tests. We planned to use a random-effects model to synthesize data if heterogeneity was present (P < 0.05 and I2 > 50%) (Shen et al., 2012). Based on this rule, pooled average sensitivity, specificity and other diagnostic parameters of pleural and blood IGRA were calculated using, respectively, a random-effects model and a fixed-effects model (Irwig et al., 1995; Vamvakas, 1998). Potential presence of publication bias was tested using funnel plots and the Egger’s test. All statistical tests were two-sided, and the threshold of significance was set at P < 0.05.

Results

Study inclusion and characteristics

Study identification and selection were outlined in Fig. 1. In the end, 15 publications of IGRA to diagnose patients with TP were eligible for inclusion (Wilkinson et al., 2005; Ariga et al., 2007; Losi et al., 2007; Baba et al., 2008; Chegou et al., 2008; Dheda et al., 2009; Lee et al., 2009; Keng et al., 2013; Ates et al., 2011; Eldin et al., 2012; Kang et al., 2012; Liu et al., 2013; Liao et al., 2014; Chung et al., 2011; Gao et al., 2012). In the studies by Dheda et al. (2009) and Kang et al. (2012), IGRA was performed in two formats: as an enzyme-linked immunosorbent spot (ELISPOT) assay, and as an enzyme-linked immunosorbent assay (ELISA). Thus, each of these publications was treated as two independent studies in our meta-analysis, giving 17 studies in our meta-analysis altogether. ELISPOT was used in seven studies (Wilkinson et al., 2005; Losi et al., 2007; Dheda et al., 2009; Lee et al., 2009; Kang et al., 2012; Liu et al., 2013; Liao et al., 2014), while ELISA was used in the remaining 10 studies (Ariga et al., 2007; Baba et al., 2008; Chegou et al., 2008; Dheda et al., 2009; Keng et al., 2013; Ates et al., 2011; Eldin et al., 2012; Kang et al., 2012; Chung et al., 2011; Gao et al., 2012). Across all studies, 17 analyses of PE (Wilkinson et al., 2005; Ariga et al., 2007; Losi et al., 2007; Baba et al., 2008; Chegou et al., 2008; Dheda et al., 2009; Lee et al., 2009; Keng et al., 2013; Ates et al., 2011; Eldin et al., 2012; Kang et al., 2012; Liu et al., 2013; Liao et al., 2014; Gao et al., 2012). and 14 analyses of blood (Wilkinson et al., 2005; Ariga et al., 2007; Losi et al., 2007; Baba et al., 2008; Chegou et al., 2008; Dheda et al., 2009; Lee et al., 2009; Ates et al., 2011; Eldin et al., 2012; Kang et al., 2012; Liu et al., 2013; Liao et al., 2014; Chung et al., 2011) were conducted. Ten studies were conducted in Asia, five in Africa, and two in Europe. Key characteristics of included studies, along with QUADAS score, were shown in Table 1.

Figure 1 Flow diagram of included and excluded studies.

Table 1 Key characteristics of the studies included in the meta-analysis.

First author	Settings	IGRA method	Samples	Test results	QUADAS score	
				TP	FP	FN	TN		
Wilkinson	UK	ELISPOT	PE	10	1	0	7	10	
			Blood	10	–	0	–		
Ariga	Japan	ELISA	PE	27	1	1	46	11	
			Blood	21	14	6	33		
Losi	Italy, Germany, Netherlands	ELISPOT	PE	19	5	1	16	10	
			Blood	18	7	2	14		
Baba	South Africa	ELISA	PE	12	2	15	4	10	
			Blood	17	0	7	6		
Chegou	South Africa	ELISA	PE	13	2	10	13	13	
			Blood	16	5	6	12		
Dheda	South Africa	ELISPOT	PE	38	8	6	9	11	
			Blood	30	7	6	9		
		ELISA	PE	23	6	19	12		
			Blood	26	4	4	9		
Lee	Taiwan	ELISPOT	PE	18	3	1	18	9	
			Blood	14	2	4	19		
Ates	Turkey	ELISA	PE	21	6	22	23	11	
			Blood	30	14	13	15		
Chuang	South Korea	ELISA	Blood	42	17	12	26	8	
Kang	South Korea	ELISPOT	PE	15	8	0	3	9	
			Blood	18	6	2	8		
		ELISA	PE	10	5	5	5		
			Blood	4	6	0	7		
Eldin	Egypt	ELISA	PE	16	3	4	15	7	
			Blood	14	7	6	11		
Gao	China	ELISA	PE	54	2	4	18	8	
Keng	Taiwan	ELISA	PE	24	2	31	57	8	
			PE	22	1	9	56		
Liu	China	ELISPOT	PE	53	2	2	41	9	
			Blood	51	10	4	33		
Liao	China	ELISPOT	PE	269	0	12	51	8	
			Blood	220	7	61	44		
Notes.

ELISPOT enzyme-linked immunosorbent spot

ELISA enzyme-linked immunosorbent assay

FN false negative

FP false positive

IGRA T-cell interferon-γ release assay

PE pleural effusion

QUADAS quality assessment for studies of diagnostic accuracy

TN true negative

TP true positive

Nine studies included at least 60 patients (Ariga et al., 2007; Chegou et al., 2008; Dheda et al., 2009; Keng et al., 2013; Ates et al., 2011; Liu et al., 2013; Liao et al., 2014; Chung et al., 2011; Gao et al., 2012). Mean sample size in the 17 analyses of pleural IGRA was 76 (range 18–332), involving a total of 806 patients with TP and 482 without TP. Mean sample size in the 14 analyses of blood IGRA was 80 (range 34–332), involving altogether 730 patients with TP and 383 without TP.

Diagnostic accuracy

In the 17 analyses of pleural IGRA, diagnostic sensitivity ranged from 0.44 to 1.0 (Fig. 2); pooled sensitivity was 0.82 (95% CI [0.79–0.85]; I2 = 92%). Specificity ranged from 0.5 to 1.0, and pooled specificity was 0.87 (95% CI [0.84–0.90]; I2 = 82.5%). Other pooled estimates of diagnostic parameters were as follows: PLR, 4.94 (95% CI [2.60–9.39]); NLR, 0.22 (95% CI [0.13–0.38]); PPV, 0.91 (95% CI [0.85–0.96]); NPV, 0.79 (95% CI [0.71–0.85]); and DOR, 28.37 (95% CI [10.53–76.4]). Chi-square values for these parameters suggested considerable heterogeneity among studies (Table 2): sensitivity, 199.86; specificity, 91.18; PLR, 129.36; NLR, 180.23; PPV, 132; NPV, 157; and DOR, 81.01 (all P < 0.001).

Figure 2 Forest plot showing estimates of sensitivity and specificity for T-cell interferon-gamma assays in pleural fluid (A) and peripheral blood (B).

Point estimates of sensitivity and specificity from each study are shown as solid circles. Error bars indicate 95% CI.

Table 2 Pooled results for accuracy of interferon-gamma assays to diagnose tuberculous pleurisy.

	Pleural effusion	Blood	
	Total	ELISPOT	ELISA	Total	ELISPOT	ELISA	
Number of study	17	7	10	14	6	8	
Sensitivity(95% CI)	0.82(0.79–0.85)	0.95(0.93–0.97)	0.65(0.60–0.70)	0.80(0.76–0.83)	0.82(0.78–0.85)	0.76(0.70–0.81)	
Heterogeneity*(P)	199.86(<0.001)	7.93(0.24)	67.65(<0.001)	19.09(0.12)	9.99(0.075)	6.17(0.52)	
Specificity(95% CI)	0.87(0.84–0.9)	0.84(0.78–0.89)	0.89(0.85–0.93)	0.70(0.65–0.75)	0.77(0.69–0.83)	0.64(0.57–0.71)	
Heterogeneity(P)	91.18(<0.001)	50.63(<0.001)	38.25(<0.001)	28.52(0.008)	12.57(0.028)	9.34(0.23)	
PLR(95% CI)	4.94(2.60–9.39)	5.62(1.65–19.14)	4.6(2.16–9.82)	2.48(1.95–3.17)	3.21(2.09–4.94)	2.00(1.63–2.45)	
Heterogeneity(P)	129.36(<0.001)	94.35(<0.001)	42.71(<0.001)	26.57(0.014)	12.18(0.03)	6.13(0.525)	
NLR(95% CI)	0.22(0.13–0.38)	0.08(0.04–0.16)	0.41(0.27–0.62)	0.30(0.24–0.37)	0.22(0.16–0.31)	0.38(0.30–0.50)	
Heterogeneity(P)	180.23(<0.001)	12.33(0.06)	48.66(<0.001)	18.02(0.16)	6.13(0.294)	5.23(0.632)	
PPV(95% CI)	0.91(0.85–0.96)	0.87(0.8–1.03)	0.96(0.68–1.32)	0.79(0.60–0.87)	0.74(0.64–0.84)	0.84(0.75–1.19)	
Heterogeneity(P)	132(<0.001)	46.52(0.03)	61.25(<0.001)	4.27(0.09)	6.79(1.22)	5.76(0.46)	
NPV(95% CI)	0.79(0.71–0.85)	0.75(0. 65–0.87)	0.84(0.7–1.41)	0.75(0.62–0.83)	0.72(0.58–0.81)	0.76(0.65–0.88)	
Heterogeneity(P)	157(<0.001)	9.68(<0.001)	16.9(<0.001)	4.54(0.07)	7.01(1.13)	11.53(0.06)	
DOR(95% CI)	28.37(10.53–76.4)	88.85(16.10–490.43)	14.10(4.56–43.54)	9.96(6.02–16.48)	19.82(11.67–33.66)	5.46(3.46–8.61)	
Heterogeneity(P)	81.01(<0.001)	23.47(0.001)	43.65(<0.001)	24.44(0.03)	4.93(0.43)	6.38(0.496)	
AUC(SEM)	0.91(0.03)	0.98(0.01)	0.84(0.08)	0.84(0.03)	0.89(0.02)	0.78(0.04)	
Notes.

* Q value.

AUC area under the curve

DOR diagnostic odds ratio

ELISPOT enzyme-linked immunosorbent spot

ELISA enzyme-linked immunosorbent assay

NLR negative likelihood ratio

PLR positive likelihood ratio

PPV positive predictive value

NPV negative predictive value

For 14 analyses of blood IGRA, diagnostic sensitivity ranged from 0.71 to 0.93 (Fig. 2), and specificity ranged from 0.56 to 1.0. Pooled estimates of the other diagnostic parameters were as follows: sensitivity, 0.8 (95% CI [0.76–0.83]; I2 = 31.9%); specificity, 0.7 (95% CI [0.65–0.75]; I2 = 54.4%); PLR, 2.48 (95% CI [1.95–3.17]); NLR, 0.3 (95% CI [0.24–0.37]); PPV, 0.79 (95% CI [0.60–0.87]); NPV, 0.75 (95% CI [0.62–0.83]); and DOR, 9.96 (95% CI [6.02–16.48]). Chi-square values for most of these parameters indicated no significant heterogeneity among studies (Table 2): sensitivity, 19.09 (P = 0.12); NLR, 18.02 (P = 0.16); PPV, 4.27 (P = 0.09); and NPV, 4.54 (P = 0.07). In contrast, chi-square values indicated significant heterogeneity for specificity (28.52), PLR (26.57) and DOR (24.44) (all P < 0.05).

This meta-analysis involved two different types of commercially available assays: ELISPOT and ELISA. The ELISPOT assay, such as the T-SPOT-TB, involves sensitizing T cells to specific M. tuberculosis antigens, such as the early secreted antigenic target 6 (ESAT-6) and culture filtrate protein 10 (CFP-10), and then measuring the IFN-γ subsequently released. ELISA, such as Quanti-FERON-TB Gold (QFN-G) or the third-generation ‘In-Tube’ (QFN-IT), measures the release of INF-γ into whole blood or PE after stimulation by ESAT-6 and CFP-10. Comparison of overall diagnostic values for ELISPOT and ELISA did not allow a conclusion about which assay type was superior (Table 2).

We assessed the overall diagnostic performance by calculating SROC curves and the corresponding AUC. The SROC curve for pleural IGRA was not positioned near the desirable upper left corner, and the point where sensitivity equals specificity (Q) was 0.84; the optimum AUC was 0.91 (Fig. 3A). The corresponding SROC curve for blood IGRA showed Q of 0.77 and AUC of 0.84 (Fig. 3B). Although neither the pleural or blood AUC was entirely satisfactory, this summary analysis suggests that pleural IGRA shows much better diagnostic performance than blood IGRA.

Figure 3 Summary receiver operating characteristic (SROC) curves for T-cell interferon-gamma assays in pleural fluid (A) and peripheral blood (B).

Solid circles represent each study included in the meta-analysis, with circle size representing the sample size in each study. The regression SROC curves summarize the overall diagnostic accuracy.

Multiple regression analysis and publication bias

The quality of the 17 studies in this meta-analysis varied considerably, with only five studies earning high QUADAS scores (≥11; Table 1). These scores were used in a meta-regression analysis to assess the effect of study quality on the relative DOR (RDOR) of IGRA for the diagnosis of TP (Table 3). Higher- and lower-quality studies did not differ significantly in RDOR for either pleural or blood IGRA (Table 3). Seven studies were performed in areas with a low tuberculosis incidence (Wilkinson et al., 2005; Ariga et al., 2007; Losi et al., 2007; Keng et al., 2013; Ates et al., 2011; Eldin et al., 2012; Kang et al., 2012) and 10 studies (eight publications) were performed in areas with a high tuberculosis incidence (Baba et al., 2008; Chegou et al., 2008; Dheda et al., 2009; Lee et al., 2009; Liu et al., 2013; Liao et al., 2014; Chung et al., 2011; Gao et al., 2012). Diagnostic accuracy of pleural IGRA depended significantly only on assay method (ELISPOT vs ELISA, P = 0.023), but not on study quality or tuberculosis incidence. Diagnostic accuracy of blood IGRA depended significantly on both assay method and tuberculosis incidence.

Table 3 Weighted meta-regression to assess the effects of study setting, IGRA method and study quality on diagnostic accuracy of IGRA.

Covariate	Number of studies	Coefficient	RDOR (95% CI)	P-value	
Pleural effusion	
QUADAS score	
≥11	5	−1.44	0.24(0.02–2.70)	0.225	
<11	12				
Setting	
Area with low TB incidence	7	0.36	1.44(0.10–21.14)	0.777	
Area with high TB incidence	10				
Method	
ELISPOT	7	−3.17	0.04(0.00–0.60)	0.023	
ELISA	10				
Peripheral blood	
QUADAS score	
≥11	5	−0.81	0.45(0.15–1.35)	0.137	
<11	9				
Setting	
Area with low TB incidence	7	1.12	3.06(1.16–8.10)	0.028	
Area with high TB incidence	7				
Method	
ELISPOT	6	−1.26	0.28(0.13–0.62)	0.0048	
ELISA	8				
Notes.

RDOR relative diagnostic odds ratio

QUADAS quality assessment for studies of diagnostic accuracy

TB tuberculosis

ELISA enzyme-linked immunosorbent assay

ELISPOT enzyme-linked immunosorbent spot

Results of the RDOR analysis were shown in Table 3.

Publication bias was analyzed by using funnel plots and the Egger’s test. Since the funnel plots for publication bias showed asymmetry (Fig. 4), Egger’s tests were performed, which confirmed significant risk of publication bias in the meta-analyses for both blood IGRA and pleural IGRA (both P < 0.001).

Figure 4 Funnel graph for assessing risk of publication bias in studies of T-cell interferon-gamma release assays in pleural fluid (A) and peripheral blood (B).

The funnel graph plots the log of the diagnostic odds ratio (DOR) against the standard error of the log of the DOR (an indicator of sample size). Solid circles represent each study inthemeta-analysis. The central lines indicate the summary DOR.

Discussion

IGRA has an advantage over conventional methods of diagnosing M. tuberculosis infection, because it is based on specific antigens, such as ESAT-6 and CFP-10, that are absent from BCG and most environmental mycobacteria. Whether this assay is suitable for diagnosing TP is controversial. In fact, Zhou et al. (2011) conducted a meta-analysis to analyze the diagnostic role of IGRA for TP. According to his inclusion criteria, only seven publications were included. Several years have passed, and some new studies have been added, so we conducted this updated meta-analysis. Our meta-analysis summarizes the available evidence on this question in an effort to provide guidance for TP diagnosis. Our results showed that the pooled sensitivities of pleural and blood IGRA were 0.82 and 0.80, respectively, and the corresponding specificities were 0.87 and 0.70. These findings, coupled with the relatively low AUC values representing overall performance, suggest that IGRA has some usefulness for diagnosing TP, but that it should be interpreted only in conjunction with conventional tests or clinical signs. Positive results from IGRA may be helpful for confirming TP, but the relatively low sensitivity makes it vulnerable to generating false negatives. Significant heterogeneity was found in sensitivity, specificity, PLR, NLR, DOR for pleural IGRA, and specificity, PLR, DOR for blood IGRA. Five studies had a higher QUADAS score (≥11). There was no significant difference between higher-quality studies and lower-quality ones.

We assessed pleural and blood IGRAs using SROC curves and DOR tests, both of which combine sensitivity and specificity. SROC curves, which are unlikely to be affected by a diagnostic threshold effect (Jones & Athanasiou, 2005), showed an optimum cut-off of 0.84 for pleural IGRA and 0.77 for blood IGRA, while the corresponding AUCs were 0.91 and 0.84, suggesting less than fully satisfactory overall accuracy. The DOR of a test is the ratio of the odds of obtaining a positive test result in the disease group to the odds of obtaining a positive test result in the no-disease group (Zhou et al., 2011). When DOR >1, higher values indicate better discriminatory test performance. We calculated a pooled DOR of 28.37 for pleural IGRA and of 9.96 for blood IGRA, suggesting that IGRA and particularly pleural IGRA may be helpful for diagnosing TP. We found higher pooled sensitivity and specificity for pleural IGRA than a previous meta-analysis (Zhou et al., 2011), which likely reflects our inclusion of more articles. Similarly we calculated a higher pooled DOR for pleural IGRA (19.0, 95% CI [4.8–75.8]) than that meta-analysis did. We conclude that pleural IGRA has better prospects than blood IGRA for widespread clinical implementation. This was possibly due to compartmentalization of antigen-specific effector T cells, which could be recruited and concentrated at the site of infection, such as pleural cavity. ESAT-6-specific, IFN-γ secreting T-cells have a 15-fold concentration in PE relative to peripheral blood in patients with TP (Wilkinson et al., 2005).

Potentially more clinically meaningful than DOR and SROC, PLR and NLR are often used as measures of diagnostic accuracy. PLR indicates how much the odds of a condition are increased by a positive test, while NLR indicates how much they are decreased by a negative test. Larger PLR means greater diagnostic accuracy, whereas a smaller NLR is better. The pooled PLR of 4.94 for pleural IGRA suggests that patients with TP have a nearly five-fold greater chance of a positive test result than patients without TP. Even though this PLR is larger than that reported in a previous meta-analysis (Zhou et al., 2011), it is still too small for clinical purposes. At the same time, we calculated a pooled NLR of 0.22 for pleural IGRA, indicating that the probability that a patient with a negative result has a 22% chance of having TP, which is not low enough to reliably rule out false negatives. The corresponding PLR and NLR for blood IGRA were even less satisfactory.

The pooled PPV for pleural IGRA was 0.91, indicating that 9% of positive results may be false positives. The NPV of pleural IGRA was 0.79, suggesting a negative rate of 21%. The corresponding values for blood IGRA were less satisfactory. Although these PPV and NPV values are higher than those reported in a recent meta-analysis (Zhou et al., 2011), they are still not as high as necessary for reliable clinical performance.

Our results are consistent with the observation that pleural and blood IGRAs give a relatively high rate of false positive test results because IGRA cannot distinguish active from latent tuberculosis (Hooper, Lee & Maskell, 2009; Dheda et al., 2009). In the present meta-analysis, we found pleural IGRA to show a lower rate of false positive results than false negative results. Previous studies showed IGRA, especially T-SPOT-TB, to be helpful in the diagnosis of latent tuberculosis (Lalvani, 2007; Pai, Zwerling & Menzies, 2008), while the overall accuracy of the technique for diagnosing TP was lower than for diagnosing latent tuberculosis (Diel et al., 2011) but higher than for diagnosing active tuberculosis (Sester et al., 2011). This dependence of diagnostic accuracy on tuberculosis form may reflect the fact that patients with latent M. tuberculosis infection live with superior immunologic function, such that smaller pathogen load can elicit an effective response to tuberculosis antigen. Another explanation is significant heterogeneity among studies. A third possible explanation is transient exposure to non-replication persistent M. tuberculosis in the pleural space of patients without PE.

Two types of IGRAs are commercially available: the ELISA-based QFT-G or QFT-IT, and the ELISPOT-based T-SPOT-TB. Although both ELISPOT and ELISA measure IFN-γ release after T cell stimulation by ESAT-6 and CFP-10, ELISPOT has been reported to be more stable and sensitive (Liebeschuetz et al., 2004). Indeed, we found the sensitivity, PLR, DOR and AUC to be higher for pleural ELISPOT than for pleural ELISA (Table 2). On the other hand, the specificity and NLR were lower for pleural ELISPOT than for pleural ELISA. In the blood-based assay, sensitivity, specificity, PLR, DOR and AUC were higher for ELISPOT than for ELISA, but NLR was lower for ELISPOT than for ELISA. Therefore, we cannot determine whether ELISPOT or ELISA shows greater overall accuracy for diagnosing TP. This requires larger studies that compare the two types of IGRAs in parallel.

The reliability of meta-analysis in general is limited by the methodological quality and heterogeneity of included studies (Petitti, 2001). Quality scoring was compiled for every study on the basis of title, introduction, methods, results and discussion. When a criterion was fulfilled, a score of 1 was given, 0 if a criterion was unclear, and −1 if a criterion was not achieved. Quality of study can be interpreted into different scores by the use of QUADAS, thus, easy to be carried out and compared. Overall the quality of study design and reporting diagnostic accuracy of most studies were good to a certain extent and five studies had a higher QUADAS score (≥11). IGRA performance was similar in higher-quality studies (QUADAS ≥11) and lower-quality ones. Pleural IGRA studies showed significant heterogeneity in meta-analyses of sensitivity, specificity, PLR, NLR and DOR. Whether the study used ELISPOT or ELISA significantly affected the diagnostic accuracy of both pleural and blood IGRAs. We also found that whether a study was performed in an area of low or high tuberculosis incidence significantly affected the accuracy of blood IGRA, but not of pleural IGRA. A previous study concluded that IGRA was more sensitive and specific than conventional methods in areas of high tuberculosis prevalence (Gao et al., 2012). This contrasts with studies in low-incidence areas showing that pleural fluid T-cells in pleural fluid respond to stimulation with ESAT-6 and CFP-10 are significantly more than do to T-cells in peripheral blood (Ariga et al., 2007; Losi et al., 2007), perhaps reflecting the fact that most patients in such areas are immunocompetent. Our observation of a differential effect of study area on the two types of IGRAs may reflect country biases in the studies examining each type of IGRA. Future studies should address this question in detail.

Theoretically, tuberculosis antigen-specific responses like the one measured by IGRA should allow clinicians to distinguish PE from alternative diagnosis and provide greater discriminatory value than non-specific inflammatory biomarkers such as unstimulated IFN-γ or adenosine deaminase (ADA). However, comparing our findings with those of previous meta-analyses (Zhou et al., 2011; Liang et al., 2008) suggests that IGRA has lower overall accuracy than either IFN-γ or ADA for diagnosing TP. In fact, one study found that combining ADA and IFN-γ to diagnose TP led to 100% specificity (Keng et al., 2013). The authors of that study were unsure why IFN-γ and ADA perform better than IGRA. Future studies should investigate this question.

Some limitations should be discussed in this meta-analysis. First, we included only studies published in PubMed and Embase, and we excluded abstracts, letters to the editor and articles written in languages other than English. This may have led to publication bias, which is indeed suggested by our funnel plots and Egger’s test. Second, only five of the 15 publications diagnosed TP based on bacteriological or histological assessment, or on the gold standard combination of both (Wilkinson et al., 2005; Ariga et al., 2007; Eldin et al., 2012; Liu et al., 2013; Gao et al., 2012). The remaining 10 publications used a mixture of bacteriological, histological or clinical assessment (Losi et al., 2007; Baba et al., 2008; Chegou et al., 2008; Dheda et al., 2009; Lee et al., 2009; Keng et al., 2013; Ates et al., 2011; Kang et al., 2012; Liao et al., 2014; Chung et al., 2011). Third, the results of this meta-analysis may be less applicable to severely immunocompromised subjects, since IGRA depends on host immunity and many studies excluded indeterminate results from analysis. This may have led to systematic error in some studies.

Conclusion

Our meta-analysis suggests that pleural IGRA shows much better diagnostic performance than blood IGRA. Pleural IGRA has potential for serving as a complementary method for diagnosing TP; but that its sub-optimal performance, cost and high turnaround time make it unsuitable as a stand-alone diagnostic tool. Better tests for the diagnosis of TP are required.

Supplemental Information

Supplemental Information 1 PRISMA checklist

PRISMA 2009 checklist.

Click here for additional data file.

Supplemental Information 2 PRISMA flow diagram

PRISMA 2009 flow diagram.

Click here for additional data file.

We are indebted to the authors of the primary studies included in this meta-analysis; without their contributions, this work would not have been possible.

Additional Information and Declarations

Competing Interests

Author Contributions

The authors declare there are no competing interests.

Cai-Shuang Pang conceived and designed the experiments, performed the experiments, analyzed the data, wrote the paper, prepared figures and/or tables, reviewed drafts of the paper.

Yong-Chun Shen conceived and designed the experiments, performed the experiments, wrote the paper, prepared figures and/or tables, reviewed drafts of the paper.

Pan-Wen Tian analyzed the data, prepared figures and/or tables.

Jing Zhu, Mei Feng and Chun Wan contributed reagents/materials/analysis tools.

Fu-Qiang Wen reviewed drafts of the paper.

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
