# Peer review of "Accuracy of the interferon-gamma release assay for the diagnosis of tuberculous pleurisy: an updated meta-analysis"

_PeerJ, doi:10.7717/peerj.951_

## Round 0.1 · original submission · Major Revisions

Thank you for submitting your manuscript. Your work has been returned from peer review. The reviewers recommended additional revisions to the manuscript before it is reconsidered for possible publication in the Journal.

We concur with the reviewers and are requesting you to revise and resubmit your manuscript. Comments from the reviewers are enclosed for your consideration in revising the text.

Carefully revise your submission and resubmit with a rebuttal letter, including point-by-point reply to the reviewers' comments and suggestions. Please note that submitting a revision of your manuscript does not guarantee acceptance.

Reviewer 1 ·

Basic reporting

In this paper, the authors tried to meta-analyzed the available evidence on the overall diagnostic performance of IGRA applied to pleural fluid and peripheral blood to investigate whether IGRA is a good method for diagnosing tuberculous pleurisy (TP). The topic is interesting, however, the result from this study is far less informative.

Experimental design

No Comments.

Validity of the findings

1. What is the novelty compared with previous meta-analysis as listed in ref such as "Sester M, Sotgiu G, Lange C, Giehl C, Girardi E, Migliori GB, Bossink A, Dheda K, Diel R, Dominguez J, Lipman M, Nemeth J, Ravn P, Winkler S, Huitric E, Sandgren A, Manissero D. 2011. Interferon-gamma release assays for the diagnosis of active tuberculosis: a systematic review and meta-analysis. Eur Respir J 37: 100-111."

2. What is the significance of this study?

3.What is the rational for using Chi-square test, multiple regression?

4. It seems that the flow of this manuscript is not well organized.

Additional comments

In this paper, the authors tried to meta-analyzed the available evidence on the overall diagnostic performance of IGRA applied to pleural fluid and peripheral blood to investigate whether IGRA is a good method for diagnosing tuberculous pleurisy (TP). The topic is interesting, however, the result from this study is far less informative.

1. What is the novelty compared with previous meta-analysis as listed in ref such as "Sester M, Sotgiu G, Lange C, Giehl C, Girardi E, Migliori GB, Bossink A, Dheda K, Diel R, Dominguez J, Lipman M, Nemeth J, Ravn P, Winkler S, Huitric E, Sandgren A, Manissero D. 2011. Interferon-gamma release assays for the diagnosis of active tuberculosis: a systematic review and meta-analysis. Eur Respir J 37: 100-111."

2.What is the significance of this study?

3. It seems that the flow of this manuscript is not well organized.

4. What is the rational for using Chi-square test, multiple regression

5. The authors should be aware of typing and grammar errors, such as in line 77: "...., ositive likelihood ratio" (should be "positive likelihood ratio")

Reviewer 2 ·

Basic reporting

No Comments

Experimental design

No Comments

Validity of the findings

No Comments

Additional comments

The diagnosis of pleural effusion can sometimes be a challenge to the clinical physicians. The article investigated the value of a new assay for tuberculous pleurisy, which is a common disease in developing countries. The investigators examined whether IGRA, a novel promising method of tuberculosis diagnosis, is an effective diagnostic method for tuberculous pleurisy. The study appears to be a worthwhile topic, especially with consideration of its potential clinical validation. However, there are some concerns regarding the manuscript.

1. The criteria of the study enrollment should be discussed in details, including the features of related study designs and the methodological quality of the studies.
3. The authors should provide a table summarizing the significant characteristics of each study in the meta-analysis.

4. The authors should evaluate the quality of the studies enrolled in the meta-analysis.
5. Regarding the discussion section, the authors should first make a short summary of the evidence quality, the heterogeneity among studies.

6. Why does pleural IGRA show much better diagnostic performance than blood IGRA? The authors should discuss the mechanism in the discussion section.
7. Pleural IGRA shows much better diagnostic performance than blood IGRA, which should be included in conclusion section due to its clinical significance.

---

## Round 0.2 · Minor Revisions

Please correct format typos as suggested by the reviewer for the purpose of publication.

Reviewer 2 ·

Basic reporting

The investigators examined whether IGRA, a novel promising method of tuberculosis diagnosis, is an effective diagnostic method for tuberculous pleurisy. Considering the worldwide burden of tuberculosis, especially in the developing countries, the study has a potential value for the clinical physician, and also for the health policy maker.

Experimental design

No Comments

Validity of the findings

No Comments

Additional comments

On the basis of amendment, the authors have made some improvements. The second edition of manuscript made some answers to the questions, which makes the paper more logical and with integrity.

Reviewer 3 ·

Basic reporting

Whether IGRA can be used to diagnose TP is controversial. In this study, the authors conduct a meta-analysis to comprehensively assess the overall accuracy of IGRA for the diagnosis of TP. The article includes sufficient introduction and backgrounds. However, there are minor errors in the format, for example, in line 116, 173, 375, 384, 468, 476 et al. Please be aware of the format and typo errors for the purpose of publication.

Experimental design

No Comments.

Validity of the findings

No comments.

Additional comments

The authors have addressed most of the questions raised by previous two reviewers point by point.

---

## Round 0.3 · accepted · Accept

The manuscipt has been accepted.